# Metal-free alcohol-directed regioselective heteroarylation of remote unactivated C(sp$^3$)–H bonds

Xinxin Wu[1], Hong Zhang[1], Nana Tang[1], Zhen Wu[1], Dongping Wang[1], Meishan Ji[1], Yan Xu[1], Min Wang[1] & Chen Zhu [1,2]

Construction of C–C bonds via alkoxy radical-mediated remote C(sp$^3$)–H functionalization is largely unexplored, as it is a formidable challenge to directly generate alkoxy radicals from alcohols due to the high bond dissociation energy (BDE) of O–H bonds. Disclosed herein is a practical and elusive metal-free alcohol-directed heteroarylation of remote unactivated C(sp$^3$)–H bonds. Phenyliodine bis(trifluoroacetate) (PIFA) is used as the only reagent to enable the coupling of alcohols and heteroaryls. Alkoxy radicals are readily generated from free alcohols under the irradiation of visible light, which trigger the regioselective hydrogen-atom transfer (HAT). A wide range of functional groups are compatible with the mild reaction conditions. Two unactivated C–H bonds are cleaved and one new C–C bond is constructed during the reaction. This protocol provides an efficient strategy for the late-stage functionalization of alcohols and heteroaryls.

---

[1] Key Laboratory of Organic Synthesis of Jiangsu Province, College of Chemistry, Chemical Engineering and Materials Science, Soochow University, 199 Ren-Ai Road, Suzhou, Jiangsu 215123, China. [2] Key Laboratory of Synthesis Chemistry of Natural Substances, Shanghai Institute of Organic Chemistry, Chinese Academy of Science, 345 Lingling Road, Shanghai 200032, China. Correspondence and requests for materials should be addressed to C.Z. (email: chzhu@suda.edu.cn)

 1

**D**irect functionalization of unactivated C(sp³)–H bonds represents one of the most intriguing and advanced technologies in synthetic chemistry but is still facing enormous challenges with reactivity and selectivity[1–4]. Radical-mediated hydrogen-atom transfer (HAT) renders an efficient entry to cleave the inert C(sp³)–H bonds that allows for subsequent substitutions[5–8]. Besides the classic Hofmann–Löffler reaction, recently new breakthroughs mediated by $N$-centered radicals have been achieved with the photoredox catalysis[9–19]. Alcohols are important and readily available chemicals. The radical-mediated late-stage functionalization of the C–H bonds of alcohols affords an ideal approach to the preparation of complex alcohol derivatives. According to the bond dissociation energy (BDE)[20–22], the C–H bonds proximal to hydroxyl group are in higher reactivity than the distal ones (Fig. 1a)[23–28]. Therefore, the selective functionalization of the less reactive remote C–H bonds in alcohols is a formidable challenge.

The HAT process triggered by alkoxy radicals has long been developed in order to gain the good chemo-/regio-selectivities[29,30]. After several decades, however, this method is still underexplored, which is mainly ascribed to the difficult homolysis of the alcoholic O–H bonds with high BDEs (ca. 105 kcal mol⁻¹) to generate alkoxy radicals from free alcohols. On the other hand, alkoxy radicals are prone to induce the $\beta$-C–C bond fragmentation under harsh reaction conditions[31–39]. Consequently, alcohols are often elaborated to other surrogates, e.g., nitrite esters[40–42], peroxy compounds[43–46], sulfonates[47–49], lead (IV) alkoxides[50–52], hypohalites[53–59], $N$-alkoxylpyridine-2-thiones[60,61], and $N$-alkoxyphthalimides[62–64], for the formation of alkoxy radicals by heat or ultraviolet irradiation. Nevertheless,

these compounds are sometimes hard to handle or synthesize. In this scenario, seeking a general and mild strategy to generate alkoxy radicals from free alcohols is highly desirable.

Recently, we disclosed a tertiary-alcohol-directed hetero-arylation of remote C(sp³)–H bonds by a sequence of HAT and intramolecular heteroaryl migrations (Fig. 1b)[65]. Alkoxy radicals were directly obtained from alcohols in the presence of iridium complex irradiated by visible light. The tertiary alcohol substrates were well designed to suit for the intramolecular mode. Soon after, Zuo et al. reported a CeCl₃-catalyzed amination of remote sp³ C–H bonds of alcohols[66]. Only primary alcohols were applied to generate alkoxy radicals (Fig. 1c). Concerning the values and ubiquity of heteroaryl moieties in drugs and bioactive molecules, the intermolecular heteroarylation of alcohols is more significant and valuable than the intramolecular mode. Herein we report our findings in the regioselectively intermolecular heteroarylation of alcohols to furnish the Minisci-type products. This reaction demonstrates a wide scope of both alcohols and heteroaryls. Notably, all types (1°, 2°, and 3°) of alcohols are apt to afford the alkoxy radicals that trigger the regioselective het-eroarylation of C(sp³)–H bonds. The metal-free conditions are mild and operationally simple, providing a practical strategy for the late-stage functionalization of alcohols and heteroaryls (Fig. 1d).

## Results

**Reaction conditions survey.** With 4-methyl quinoline **1a** and $n$-pentanol **2a** as model substrates, we set about investigating the reaction parameters (Table 1). After a brief survey, it was found

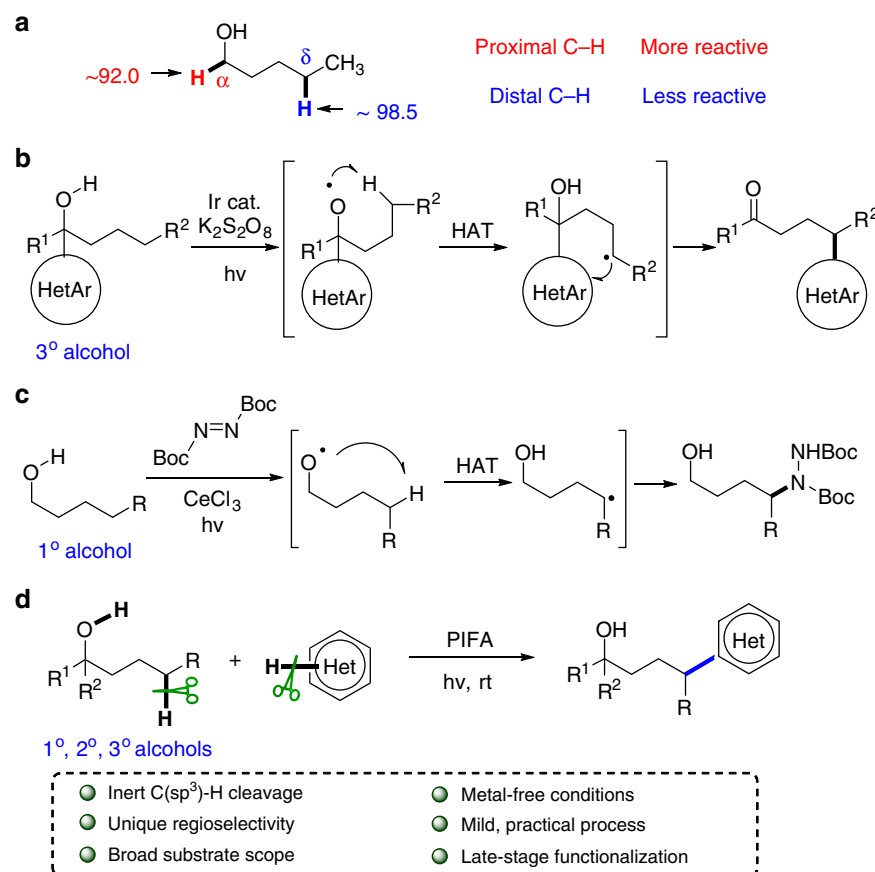

**Fig. 1** Free alcohol-directed functionalization of remote C(sp³)–H bonds. **a** BDEs (kcal mol⁻¹) of C(sp³)–H bonds in alcohols. **b** Intramolecular heteroarylation of tertiary alcohols. **c** Intermolecular amination of primary alcohols. **d** Intermolecular heteroarylation of alcohols

**Table 1 Reaction parameters survey**

| Entry | HI (x) | hv | Solvent | Yield (%)[a] |
|---|---|---|---|---|
| 1 | PIFA (2.0) | 30 W blue LEDs | DCM | 74 |
| 2 | PIFA (2.0) | 30 W CFL bulb | DCM | 57 |
| 3 | PIFA (2.0) | 30 W green LEDs, or in dark | DCM | 0 |
| 4 | PIFA (2.0) | In dark, 50 °C | DCM | 36 |
| 5 | PIFA (2.5) | 30 W blue LEDs | DCM | 76 |
| 6 | PIFA (3.0) | 30 W blue LEDs | DCM | <5 |
| 7 | F$_5$-PIFA (2.5) | 30 W blue LEDs | DCM | 39 |
| 8 | PIDA (2.5) or BI-OH (2.5) or BI-OAc (2.5) | 30 W blue LEDs | DCM | <5 |
| 9 | PIFA (2.5) | 30 W blue LEDs | DCE | 71 |
| 10 | PIFA (2.5) | 30 W blue LEDs | MeCN | 70 |
| 11 | PIFA (2.5) | 30 W blue LEDs | CHCl$_3$ | 38 |
| 12 | PIFA (2.5) | 30 W blue LEDs | PhCF$_3$ | 53 |
| 13 | PIFA (2.5) | 30 W blue LEDs | DMF | <5 |
| 14 | PIFA (2.5) | 30 W blue LEDs | DMSO | 0 |
| 15 | PIDA (2.5), I$_2$ (1.0) | 30 W blue LEDs | DCM | 0 |
| 16 | PIFA (2.3) | 100 W blue LEDs | DCM | 90 |
| 17[b] | PIFA (2.3) | 100 W blue LEDs | DCM | 79 |
| 18[c] | PIFA (2.3) | 100 W blue LEDs | DCM | 70 |
| 19[d] | PIFA (2.3) | 100 W blue LEDs | DCM | 43 |
| 20[e] | PIFA (2.3) | 100 W blue LEDs | DCM | 30 |

Reaction conditions: **1a** (0.4 mmol), **2a** (2.0 mmol, 5 equiv.), and **HI 1–5** (as shown) in solvent (2.0 mL), rt, visible-light irradiation
[a]Yields of isolated products
[b]**2a** (1.6 mmol, 4 equiv.)
[c]**2a** (1.2 mmol, 3 equiv.)
[d]**2a** (0.8 mmol, 2 equiv.)
[e]**2a** (0.4 mmol, 1.0 equiv.), PIFA (0.3 mmol, 0.75 equiv., added in three batches)

that the only use of stoichiometric amounts of hypervalent iodine (III) reagent phenyliodine bis(trifluoroacetate) (PIFA) could promote the reaction under the irradiation of blue light-emitting diodes (LEDs) (entry 1). While varying the light source to compact fluorescent light (CFL) bulb decreased the yield (entry 2), the reaction did not proceed with the use of green LEDs or in dark (entry 3). The reaction also took place at 50 °C without light irradiation albeit in a low yield (entry 4), suggesting that this is not a photocatalytic process and the light may just input energy into the reaction. Increasing the amount of PIFA slightly improved the outcome (entry 5), but using too much PIFA significantly inhibited the reaction (entry 6). Other hypervalent iodine(III) reagents were also examined. Surprisingly, the use of (diacetoxyiodo)benzene (PIDA) almost turned off the reaction (entry 8). The use of 1,2-dichloroethane (DCE) or MeCN instead of dichloromethane (DCM) delivered comparable yields (entries 9 and 10), but other solvents did not work efficiently (entries 11–14). The Suárez's conditions (PIDA and I$_2$)[56–59], which were often applied to the intramolecular cyclization reactions via the generation of alkoxy radicals from hypohalite intermediates, were

not suitable for our reaction (entry 15). This result clearly illustrated that the current reaction underwent a non-trivial pathway rather than the formation of hypohalite intermediates. The yield was further elevated to 90% by treating the reaction with high-intensive blue LEDs (entry 16). However, it was found that the reaction yield was going down along with reducing the amount of alcohols (entries 17–20).

**Scope of heteroaryls and alcohols**. With the optimized reaction conditions in hand, we turned to examine the generality of protocol (Fig. 2). Firstly, a variety of heteroaryls were tested. The electronic properties of heteroaryls did not have much influence on the reaction, as both electron-donating (e.g., Me, OMe) and electron-withdrawing (e.g., Cl, CN, CO$_2$Et) groups were well tolerated (**3b**–**3f**). While the reaction could take place at both the *ortho-* and *para-*positions of quinoline (**3g**) or pyridine (**3k**), the reaction of isoquinoline solely preferred the 1-position (**3h**–**3j**). The example of **3i** was noteworthy, since the presence of bromide reserved a platform for the late-stage

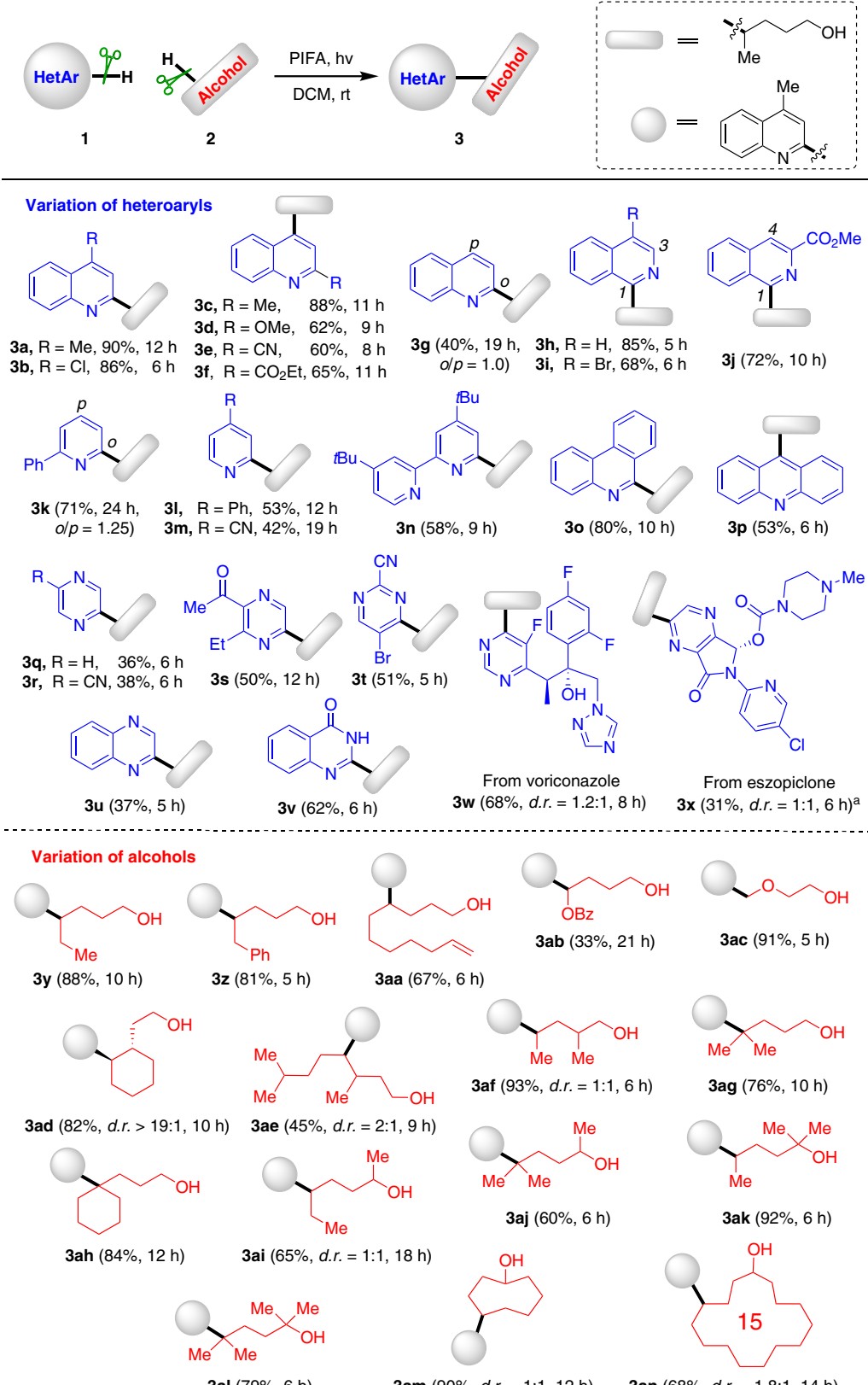

**Fig. 2** Scope of heteroaryls and alcohols. Reaction conditions: heteroaryl **1** (0.4 mmol), alcohol **2** (2.0 mmol), and PIFA (0.92 mmol) in DCM (2 mL), irradiated by 100 W blue LEDs, rt. Yields of isolated products are given. [a]1.5 equiv. TFA was added. PIFA: phenyliodine bis(trifluoroacetate)

products manipulation by cross-coupling reactions. The conversion of N,N-bidentate ligand dtbpy to **3n** provided a valuable tactic for the ligand modification. Other heteroaryls, such as phenanthridine (**3o**), acridine (**3p**), pyrazine (**3q**–**3s**), pyrimidine (**3t**), quinoxaline (**3u**), and 4-hydroxyquinazoline (**3v**) were also proved to be suitable substrates, indicating a broad scope of heteroaryls. While the pyrazine product **3r** was formed along with another regio-isomer (~10%), the product **3s** was delivered in a unique regioselectivity, which was determined by the heteronuclear multiple bond correlation (HMBC) analysis. Significantly, the method could be applied to complex molecules, such as voriconazole and eszopiclone, affording the corresponding products in good regioselectivities (**3w** and **3x**). Next, a number of alcohols were investigated. The alkoxy radical-mediated 1,5-HAT exclusively occurred even in the presence of more reactive benzylic C–H bonds (**3z**), showing the outstanding regioselective control. It might be attributed to that the 1,5-HAT via six-membered cyclic transition state is more kinetically favorable in this reaction. A variety of Minisci-type products were readily furnished. It should be noted that the current Minisci reactions involving C(sp³)–H activation are largely dependent upon the inherent BDEs of C–H bonds, and scarcely discuss about the regioselective control[67–70]. The olefin moiety, which is generally susceptive to radical process,

remained intact in the reaction (**3aa**). The C–H bonds adjacent to heteroatoms were also readily functionalized (**3ab** and **3ac**). Remarkably, the reaction with cyclic C–H bonds proceeded stereoselectively, leading to the thermodynamically favored *trans*-product (**3ad**). In contrast, the reaction with linear C–H bonds delivered the diastereomer mixtures in a 2:1 or 1:1 ratio (**3ae** and **3af**). In addition to primary and secondary C–H bonds, the congested tertiary C–H bonds were also readily transformed (**3ag** and **3ah**), forming the new quaternary all-carbon centers. The applicability was further spread from primary to secondary and tertiary alcohols. Both of them were suitable precursors for the generation of alkoxy radicals to accomplish the distal C–H bond heteroarylation (**3ai**–**3al**). This method provides an efficient approach for alcohol derivatization. For instance, the heteroaryl groups could be directly introduced to the δ-position of cycloalkanols (**3am** and **3an**). The regioselectivities were unambiguously determined by the HMBC experiments.

**Mechanistic studies**. To shed light on the mechanistic pathways, a set of experiments were carried out. Replacement of the hydroxyl group by ether entirely inhibited the reaction, verifying that the reaction was enabled by free alcohols (Fig. 3a).

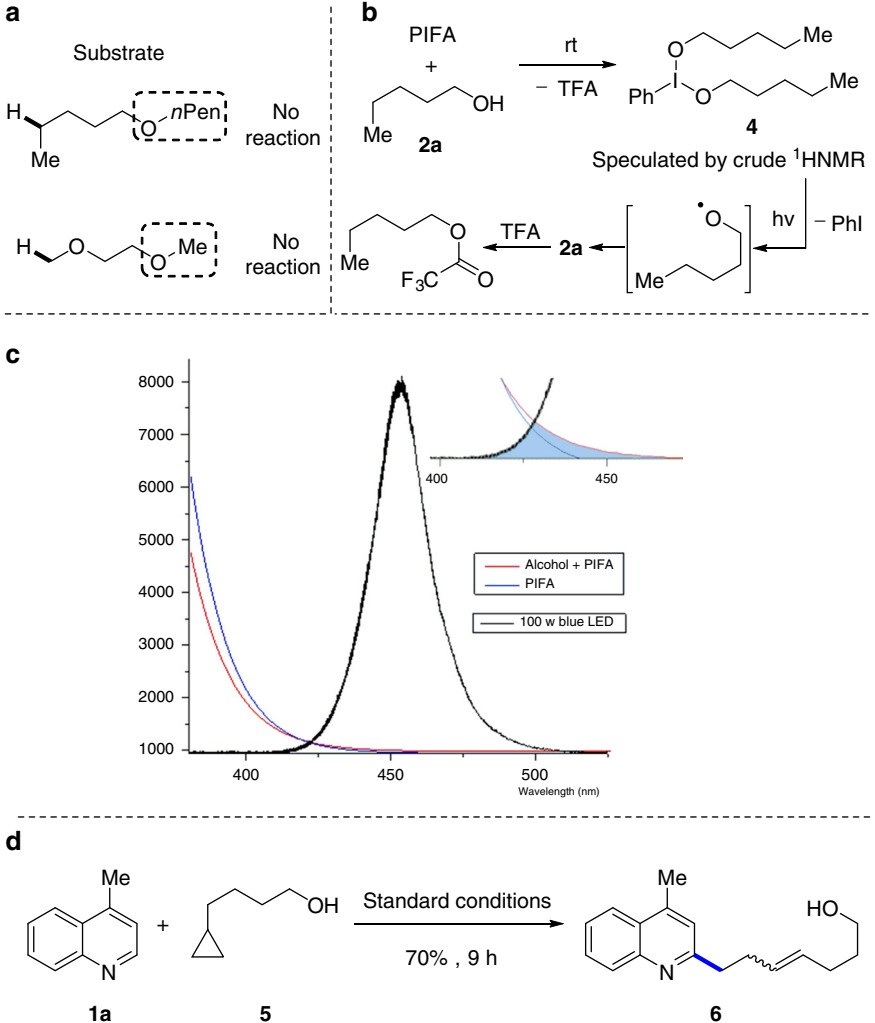

**Fig. 3** Mechanistic studies. **a** Reaction with ethers instead of alcohols. **b** NMR studies for mechanistic investigation. **c** Absorption spectra (PIFA and **4**) and emission spectra (blue LEDs). **d** Radical clock experiment. PIFA: phenyliodine bis(trifluoroacetate)

**Fig. 4** Plausible reaction mechanism. The proposed pathways for the generation of alkoxy radical from alcohol and the subsequent Minisci-type reaction

Mixing **2a** with PIFA in *d*-chloroform immediately led to a metastable intermediate **4**, which was speculated by crude $^1$H NMR. Then irradiating the mixture with blue LEDs resulted in the formation of **2a** and trifluoroacetate thereof presumably via the alkoxy radical intermediate (Fig. 3b). The intermediate **4** displayed weak absorption from 420 to 450 nm, suggesting the possibility of energy transfer from blue LEDs to **4** (Fig. 3c). Finally, the radical clock experiment unambiguously provided support for the proposed radical pathway (Fig. 3d). The PIFA-promoted reaction of **1a** with **5** afforded the ring-opened product **6** in 70% yield as a 4:1 mixture of E and Z isomers.

The proposed mechanism was depicted on the basis of experimental observations (Fig. 4). Initially, the mixture of **2a** and PIFA results in the dialkoxyiodo benzene **4**. Homolysis of **4** induced by visible-light irradiation leads to the alkoxy radical **I** that triggers the subsequent 1,5-HAT to generate the alkyl radical **II**. Meanwhile, PhI and iodanyl radical are cogenerated in the reaction. Nucleophilic addition of **II** to the quinoline salt **III** affords the radical cation **IV**, which is then single-electron oxidized by excess PIFA or the in situ formed iodanyl radical to afford the final product **3a**.

## Discussion

In summary, we have described a practical and metal-free protocol of alcohol-directed remote C(sp$^3$)–H functionalization. The combinational use of PIFA and visible-light irradiation offers a non-trivial and mild tactic for the direct generation of alkoxy radicals from free alcohols. This strategy is expected to significantly facilitate the alkoxy radical-mediated transformations. A vast array of heteroaryls and alcohols have proven to be suitable substrates. The protocol makes a complement to the classic Minisci reactions, and may find wide use in medicinal synthesis owing to the easy operation and metal-free conditions.

## Methods

**General procedure for heteroarylation of remote C(sp$^3$)–H bonds**. Heteroaryl **1** (0.4 mmol) and alcohol **2** (2.0 mmol) were loaded in a reaction vial, which was subjected to evacuation/flushing with N$_2$ three times. Then DCM (2.0 mL) followed by PIFA (0.92 mmol) was added to the mixture. The reaction was irradiated with 100 W blue LEDs and kept at room temperature (rt) under fan cooling. After the reaction completion monitored by TLC, the mixture was quenched by addition of aq. KOH until pH > 8 and then extracted with ethyl acetate (3 × 10 mL). The combined organic extracts were washed by brine, dried over Na$_2$SO$_4$, filtered, concentrated, and purified by flash column chromatography on silica gel (eluent: ethyl acetate/petroleum ether) to give the desired product **3**.

**Data availability**. The authors declare that all other data supporting the findings of this study are available within the article and Supplementary Information files, and also are available from the corresponding author on reasonable request.

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

## Acknowledgements

C.Z. is grateful for the financial support from Soochow University, the National Natural Science Foundation of China (21722205), the Project of Scientific and Technologic Infrastructure of Suzhou (SZS201708), and the Priority Academic Program Development of Jiangsu Higher Education Institutions (PAPD).

## Author contributions

X.W. and C.Z. conceived and designed the experiments; X.W. carried out most of the experiments; X.W., H.Z., N.T., Z.W., D.W., M.J., Y.X., M.W. and C.Z. analyzed the data; C.Z. wrote the paper; C.Z. directed the project.

## Additional information

**Competing interests:** The authors declare no competing interests.

