## [Peer Review File · Nature Communications]

Reviewer #1 (Remarks to the Author):

This manuscript reported by Zhu and coworkers disclosed a novel and practical alcohol-directed heteroarylation of remote unactivated C(sp³)-H bonds. In this visible light induced metal-free transformation, alkoxy radicals are readily generated from free alcohols without pre-functionalization. Various functional groups are compatible and mechanism studies were conducted. This is a powerful strategy for the synthesis of various heterocycle substituted alcohols.

The Barton reaction have long been proven to be powerful tools in organic synthesis. The unique reactivity and selectivity rendered by alkoxy radicals outcompetes most of the C-H functionalizations developed nowadays. However, the Barton reaction generally require pre-functionalization, the direct generation of alkoxy radicals under mild conditions is still in high demanding. Although Zuo et al. reported a CeCl₃-catalyzed amination of remote sp³ C-H bonds of alcohols recently [J. Am. Chem. Soc. 140, 1612-1616 (2018)], this novel transformation provided a different way for the direct generation of alkoxy radicals under metal free conditions. The mechanism is instructive in organic radical transformations.

Overall, I think this is a very nice chemistry for publication in Nature Communications after minor revisions:

1. One of the attracting points of this reaction is the high selectivity. I am curious that for substrate 3z, did the authors detect any oxidation products at the benzylic C-H bonds?
2. The use of alcohols is 5.0 equiv, had the authors tried to use 1 equiv alcohols as substrates? What is the result?
3. In mechanistic studies in Fig. 1B, 2a could react with in-situ generated TFA to afford trifluoroacetate. Had the authors detect the trifluoroacetate under the existence of heterocycles 1?
4. PIFA is crucial in this reaction. A key byproduct is PhI based on proposed mechanism. The detection of PhI would be more convincing for the effect of PIFA proposed by the authors.

Reviewer #2 (Remarks to the Author):

Wu et al. disclose in elegant work alcohol-directed regioselective heteroarylation. Although the individual steps of the cascade are established, the overall sequence represents an interesting and highly useful process. Radical C-H functionalization is a highly active and important research area, although first reports in that area (such as the HLF-reaction) were disclosed a long time ago. In general, reactive amidyl radicals (HLF-reaction) or alkoxy radicals (Barton nitrite ester reaction) are generated in situ, which then undergo intramolecular 1,5 HAT to give C-centered radicals which can finally be trapped by various reagents. The current method uses PIFA for alkoxy radical generation from the corresponding alcohols (Suarez variant). The C-radical generated after 1,5 HAT is then trapped intermolecularly by protonated heteroarenes (Minisci reaction). Elegantly, the acid (TfOH) is generated in situ from PIFA. Typical regioselectivities for Minisci-type heteroarylations are observed. Yields are moderate to very good and the scope is well documented. The cascade works for primary, secondary and tertiary alcohols. There are few examples of regioselective radical C-H functionalization comprising C-C bond formation. Remote C-H heteroarylation is not well explored (one example is the authors previous work, as cited in the introduction). This contrasts the many examples on remote C-X bond forming processes. Therefore, I rate the quality of the submitted paper, that is written well, as very high and support publication in Nature Communications subject to modifications:

1) A problem of the method lies in the reagent stoichiometry. By looking at the title, the alcohol should be the limiting reaction component. However, authors use a 5-fold excess of the alcohol. Considering heteroarylation of a sophisticated alcohol this is not acceptable. For complex heteroarenes (see product 3x) this is ok. Therefore, it would be great if authors can provide a protocol B (method B) considering the alcohol as limiting reaction component.

2) Mechanism: I(III)-species 4 is considered as the intermediate, which upon I-O bond homolysis affords the alkoxy radical. At the same time an iodanyl radical is cogenerated. The fate of that iodanyl radical is not discussed. Note that the iodanyl radical is also an SET-oxidant or it can also abstract H-atoms. The latter reactivity is not likely herein because authors get a highly regioselective radical formation. The iodanyl radical can also oxidize radical cation IV in my eyes.

3) Is regioselectivity for the formation of 3z perfect? The 1,6-HAT would deliver a stabilized secondary benzylic radical. Please comment on that point.

Response Letter

Dear Dr. Giovanni Bottari,

Thanks for your positive reply. We herein respond to the comments from both reviewers point-to-point. The manuscript is slightly modified according to the reviewer's suggestions, and the changes are highlighted. We hope you will be satisfied with this version.

Response to Reviewer 1

1. "One of the attracting points of this reaction is the high selectivity. I am curious that for substrate 3z, did the authors detect any oxidation products at the benzylic C-H bonds?"

Response: The product 3z was generated in high yield via 1,5-HAT (six-membered cyclic transition state). The benzylic oxidation byproduct formed via 1,6-HAT was not detected by NMR analysis. It might be attributed to that 1,5-HAT is more kinetically favorable in this reaction.

2. "The use of alcohols is 5.0 equiv, had the authors tried to use 1 equiv alcohols as substrates? What is the result?"

Response: We have carefully investigated the amount of alcohols. The yield of desired product was dropped to 30% yield while 1 equiv. of alcohol was used. For details, see the revised Table 1.

3. "In mechanistic studies in Fig. 1B, 2a could react with in-situ generated TFA to afford trifluoroacetate. Had the authors detected the trifluoroacetate under the existence of heterocycles 1?"

Response: In the presence of heterocycles 1, a few amount of trifluoroacetate can be detected by crude NMR.

4. "PIFA is crucial in this reaction. A key byproduct is PhI based on proposed mechanism. The detection of PhI would be more convincing for the effect of PIFA proposed by the authors."

Response: The generation of PhI during the reaction can be clearly verified by either TLC or crude NMR. The related NMR spectra are provided below.

¹H NMR for PhI (CD₃CN):

7.815
7.813
7.794
7.769
7.750
7.749
7.732
7.730
7.728
7.731
7.722
7.702

Parameter	Value
Origin	Bruker BioSpin GmbH
Solvent	CD3CN
Temperature	298.2
Number of Scans	2
Spectrometer Frequency	400.13
Nucleus	1H

¹³C NMR for PhI (CD₃CN):

Parameter	Value
Origin	Bruker BioSpin GmbH
Solvent	CD3CN
Temperature	298.2
Number of Scans	20
Spectrometer Frequency	100.61
Nucleus	13C

136.951
130.028
127.248
116.824

¹H NMR for crude reaction (CD₃CN):

Parameter	Value
Origin	Bruker BioSpin GmbH
Solvent	CD ₃ CN
Temperature	298.1
Number of Scans	20
Spectrometer Frequency	100.61
Nucleus	¹³ C

136.913
129.980
127.197
116.752

¹³C NMR for crude reaction (CD₃CN):

7.725
7.722
7.704
7.701
7.390
7.387
7.384
7.369
7.352
7.350
7.347
7.164
7.159
7.155
7.139
7.134
7.120

0.000

Parameter	Value
Origin	Bruker BioSpin GmbH
Solvent	CD ₃ CN
Temperature	298.1
Number of Scans	2
Spectrometer Frequency	400.13
Nucleus	¹ H

Response to Reviewer 2

1. "A problem of the method lies in the reagent stoichiometry. By looking at the title, the alcohol should be the limiting reaction component. However, authors use a 5-fold excess of the alcohol. Considering heteroarylation of a sophisticated alcohol this is not acceptable. For complex heteroarenes (see product 3x) this is ok. Therefore, it would be great if authors can provide a protocol B (method B) considering the alcohol as limiting reaction component."

Response: Thanks for the suggestion. We have carefully investigated the amount of alcohols. The results are included in the revised Table 1. However, it was found that the yield of product was going down along with reducing the amount of alcohols. It is reasonable as the alkoxy radical is highly reactive species. Besides the 1,5-HAT process in this reaction, the alkoxy radical could also be terminated by the beta-scission of C-C bond or other HAT processes. So we need excess alcohols to ensure the satisfactory yields. In the case of a sophisticated alcohol, it is worthy noting that a part of the excess alcohols could be recovered. Moreover, the use of excess heteroarenes will significantly inhibit the reaction.

An example showed below is the reaction of **1a** with the reduced amounts (1 equiv.) of alcohol to afford product **3ak**. The yield of product is apparently much lower than that under the standard reaction conditions (92%). Thus, the amount of alcohols is crucial to the reaction outcome currently.

2. "Mechanism: I(III)-species 4 is considered as the intermediate, which upon I-O bond homolysis affords the alkoxy radical. At the same time an iodanyl radical is cogenerated. The fate of that iodanyl radical is not discussed. Note that the iodanyl radical is also an SET-oxidant or it can also abstract H-atoms. The latter reactivity is not likely herein because authors get a highly regioselective radical formation. The iodanyl radical can also oxidize radical cation IV in my eyes."

Response: Thanks for the valuable suggestion. Indeed, the iodanyl radical could also oxidize radical cation IV if it existed. We have revised Fig. 4 and the relative description accordingly.

3. "Is regioselectivity for the formation of 3z perfect? The 1,6-HAT would deliver a stabilized secondary benzylic radical. Please comment on that point."

Response: Yes, the formation of 3z was regioselectively proceeded in high yield. The benzylic substituted product generated via 1,6-HAT was not detected by NMR analysis. It might be attributed to that the 1,5-HAT via six-membered cyclic transition state is more kinetically favorable in this reaction. We have commented it in the manuscript.

Thanks for your kind consideration.

Sincerely Yours,
Chen Zhu, Ph.D., Professor
Soochow University, China

Reviewer #1 (Remarks to the Author):

My questions were addressed in the revised manuscript. I believe the quality of this current manuscript has been improved, which is recommended publish at this version.

Reviewer #2 (Remarks to the Author):

Authors addressed all the concerns I raised in my referee report. Although the stoichiometry of the reaction is not perfect at all, I feel that the conceptual novelty of the remote arylation warrants publication in Nature Communications. The other two issues I raised were addressed and the paper further improved in quality. I now support publication as it is.